# Landslide Susceptibility Mapping Based on Multitemporal Remote Sensing Image Change Detection and Multiexponential Band Math

Xianyu Yu [1,2,*], Yang Xia [1], Jianguo Zhou [1,2] and Weiwei Jiang [1,2]

1  School of Civil Engineering, Architecture and Environment, Hubei University of Technology, Wuhan 430068, China
2  Innovation Demonstration Base of Ecological Environment Geotechnical and Ecological Restoration of Rivers and Lakes, Hubei University of Technology, Wuhan 430068, China
*  Correspondence: yuxianyu@hbut.edu.cn

**Abstract:** Landslides pose a great threat to the safety of people's lives and property within disaster areas. In this study, the Zigui to Badong section of the Three Gorges Reservoir is used as the study area, and the land use (LU), land use change (LUC) and band math (band) factors from 2016–2020 along with six selected commonly used factors are used to form a land use factor combination (LUFC), land use change factor combination (LUCFC) and band math factor combination (BMFC). An artificial neural network (ANN), a support vector machine (SVM) and a convolutional neural network (CNN) are chosen as the three models for landslide susceptibility mapping (LSM). The results show that the BMFC is generally better than the LUFC and the LUCFC. For the validation set, the highest simple ranking scores for the three models were obtained for the BMFC (37.2, 32.8 and 39.2), followed by the LUFC (28, 26.6 and 31.8) and the LUCFC (26.8, 28.6 and 20); that is, the band-based predictions are better than those based on the LU and LUC, and the CNN model provides the best prediction ability. According to the four groups of experimental results with ANNs, compared with LU and LUC, band is easier to access, yields higher predictive performance, and provides stronger stability. Thus, band can replace LU and LUC to a certain extent and provide support for automatic and real-time landslide monitoring.

**Keywords:** landslide susceptibility mapping (LSM); land use/land use change (LU/LUC) factors; band math (band) factor; artificial neural network (ANN); support vector machine (SVM); convolutional neural network (CNN)

## 1. Introduction

Landslides are natural geological disasters that can cover large areas, cause serious harm and have complex conditions; notably, they are a manifestation of geomorphological evolution [1]. Located in the eastern part of Asia, China has complex geological structures and is a country of extensive human activities that has suffered from landslides for a long time [2–4]. According to data from the National Bureau of Statistics and the National Geological Disaster Bulletin, in the past ten years, from 2012 to 2020, a total of 84,846 geological disasters (landslides, collapses, mudslides, ground subsidence, etc.), including 59,140 landslides, which account for 69.7% of all geological disasters, occurred in China. To date, there have been a total of 4421 casualties from geological disasters, resulting in a direct economic loss of RMB 41.26 billion [5], as shown in Figure 1. Scholars have conducted qualitative, semiquantitative, semiqualitative and qualitative research on landslide susceptibility mapping (LSM) and have developed many applicable models. However, relatively little research has been conducted on landslide factors, especially with the gradual deepening of the understanding of landslide phenomena, and it has become necessary to add land use (LU) factors to LSM studies. The accuracy of LU factors is extremely dependent on the

quality of remote sensing (RS) images, the choice of classifier, and the subjective judgment of the operator. Thus, LU factors can be unstable, which leads to unstable land use change (LUC) factors. Therefore, this paper focuses on the influence of the band math (band) factor, which is relatively stable and not subjectively influenced by the operator, on LSM.

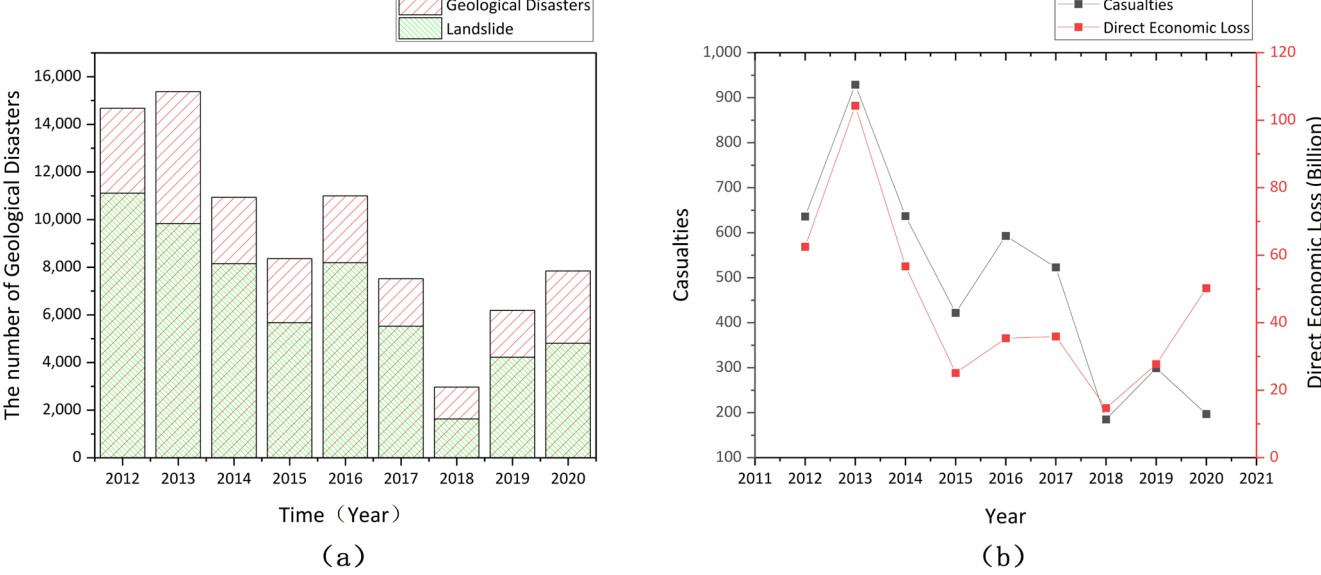

**Figure 1.** (**a**) The number of geological disasters and number of landslides in China from 2012 to 2020. (**b**) The number of casualties and direct economic losses in China from 2012 to 2020.

Scholars around the world use qualitative or quantitative methods for LSM analysis. Qualitative methods rely on expert opinions and are highly subjective, while quantitative methods focus on the relationship between various factors and landslide occurrence [6,7]. With the development of geographic information systems (GISs) and RS technologies in recent years, as well as the continuous innovation and optimization of computer software and hardware, machine learning algorithms (MLAs) have been widely used in LSM and use different MLA models, such as logistic regression (LR) [8–10], artificial neural network (ANN) [11–13], support vector machine (SVM) [14–18], Bayesian algorithm [19–21], random forest (RF) [22,23], extreme gradient boosting (XGBoost) [24,25], k-fold cross-validation (CV) [26,27] and ensemble learning [28] models. With the growth of deep learning, convolutional neural networks (CNNs) have also been applied to LSM and provide good predictive ability [29–32]. Although the above models have displayed good applicability in the field of LSM, there is no single model that works best in all scenarios, so the performance of each model needs to be compared in different situations [33]. The application of various models in LSM is relatively mature, but research on landslide factors is relatively rare.

In recent years, with the gradual deepening of the understanding of the mechanisms of landslide occurrence, scholars have gradually given more attention to LSM factors and, in particular, have obtained a more profound understanding of the crucial role of human engineering activities in the occurrence of landslides [34–36]; thus, LU factors have received some attention from researchers [37–40]. Notably, the time factor was added to the LU factor obtain the LUC factor, which has been gradually recognized by scholars and used in LSM. Soma and Kubota et al. [41–43] used the LUC factor for LSM in multiple studies and believed that it is an important factor in LSM. Meneses et al. [44] evaluated the impacts of LU and LUC factors on landslide susceptibility zonation (LSZ) in a road network in the Zêzere watershed in Portugal. Chen et al. [45] selected Zhushan town, Xuan'en, as the study area and discussed the influence of LU and LUC factors on LSM. Although the importance of LU/LUC factors in LSM has been gradually recognized, the LU/LUC factors obtained from field geological surveys are characterized by poor timeliness. Moreover, it is difficult to obtain high classification accuracy using a traditional RS classification algorithm

based on statistics when the ground environment is complex [46], and the LU/LUC factors obtained by using various classifiers through RS images rely heavily on operator experience, professional knowledge and subjective judgment and often lack accuracy. With the rapid development of RS technologies, various indices have been obtained through the band math of RS image data and used by scholars in LSM research. Ma et al. [47] used QuickBird RS images along the Yalu River estuary to combine principal component analysis (PCA), the maximum likelihood method and band math, introduced the concept of stratification and proposed a high-resolution land use RS classification method. Hassan [48] used the normalized difference vegetation index (NDVI), normalized difference water index (NDWI) and normalized difference building index (NDBI) to represent vegetation, water bodies and built-up land categories, respectively, and applied unsupervised classification results to show that the spectral features of the three land categories were easier to distinguish in the obtained images than in the original images (in Arabic). Jawak et al. [49] used the mean values of different spectral bands and spectral metrics such as the NDVI, NDWI and NDBI for classification to determine the difference thresholds between different LU/LUC categories. Yang [50] used the ENVI as a platform, applied the band math tool to obtain the NDVI with outliers removed and established a mixed-pixel decomposition model based on the principle of pixel dichotomy to estimate and optimize vegetation coverage. RS images provide various indices through band math and spectral features, and they can be used to simplify manual and tedious operations and reduce the influence of operator subjectivity on LU/LUC division to a considerable extent.

In this paper, the Zigui to Badong section of Three Gorges Reservoir was selected as the study area, and Landsat 8 data from 2015 to 2020 were used. The Landsat 8 satellites were launched in 2013, and compared with other Landsat satellites, the RS images obtained with the Landsat 8 satellites provide better image quality. The years 2015–2020 were relatively late in the Landsat series, which is of guiding significance for the prediction of future landslides. The LU, LUC and band factors were obtained by manipulation of Landsat 8 images. Then, the LU, LUC and band factors were combined with six commonly used factors (altitude, slope, slope aspect, rainfall, terrain wetness index (TWI) and lithology), and the resulting factor combinations were established using three models (ANN, SVM and CNN) for LSM. The receiver operating characteristic (ROC) curve, specific category accuracy and five statistical methods were used to evaluate and analyze the results. Finally, a simple ranking method was used to comprehensively evaluate the prediction performance, and an additional four sets of experiments were conducted with the ANN model to evaluate the LSM results of three different factor combinations to improve the scientific basis, accuracy and timeliness of LSM.

## 2. Study Area, Data and Software Introduction

### 2.1. Study Area

The geographic coordinates of the study area are $110°18'$–$110°52'$ E longitude and $30°01'$–$30°56'$ N latitude, extending from Badong County along the Yangtze River to Zigui County, which includes two mountain ranges, resulting in obvious undulating terrain in the study area [51], as shown in Figure 2. The main geological lithologies in the study area include sandstone, shale and mudstone. The Yangtze River and its main stream run through the entire study area from west to east, forming a complex water system network located in a subtropical monsoon climate zone. The four seasons are distinct, and rainfall is abundant. The average number of rainstorm days is approximately seven per year, and the maximum daily rainfall reaches 200–250 mm. Heavy rain is one of the important factors affecting the occurrence of geological disasters such as landslides and debris flows in the study area [52]. Urbanization and infrastructure development activities have further aggravated the occurrence of geological disasters in the study area [53]. The study area has complex geological and topographic structures, high mountains, deep valleys, large terrain slopes and little natural vegetation, and it is affected by heavy rainstorms and frequent human engineering activities. These factors result in a high probability of landslides in

China. Reservoir-type landslides are the most common geological disasters in the Three Gorges Reservoir Area (TGRA), with the characteristics of group occurrence, simultaneity, explosiveness and large extents [54].

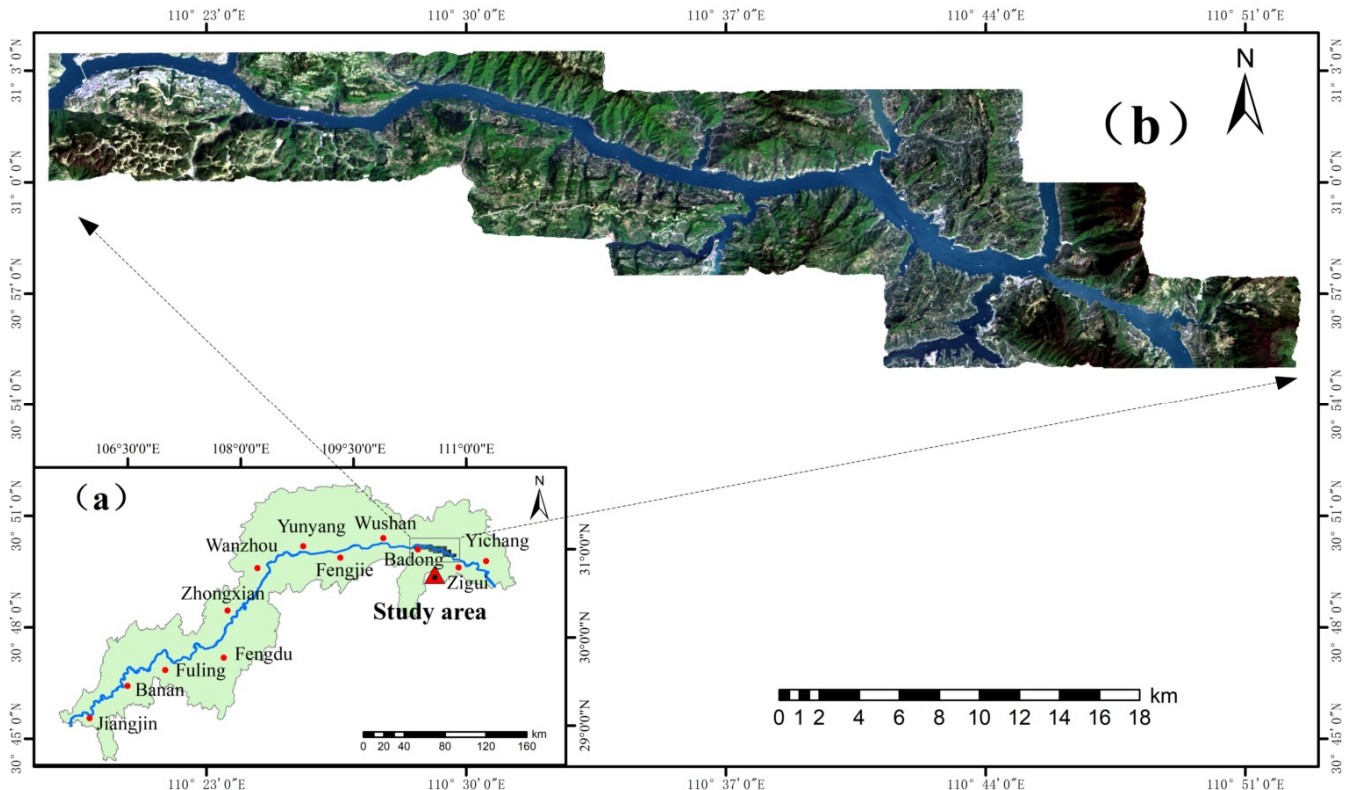

**Figure 2.** (**a**) Map of the TGRA; (**b**) geographical location of the study area.

## 2.2. Datasets

A DEM was obtained from the ASTER GDEM V2 data (https://asterweb.jpl.nasa.gov/gdem.asp) (accessed on 1 December 2021) provided by the National Aeronautics and Space Administration (NASA). The obtained DEM was used to generate altitude, slope, aspect and TWI information. Lithology and landslide hazard information were obtained from geological maps and landslide hazard maps provided by the China Geological Survey (https://www.cgs.gov.cn/) (accessed on 2 December 2021). Rainfall data were obtained from the China Meteorological Administration (http://www.cma.gov.cn/) (accessed on 2 December 2021). RS data were acquired from Landsat 8 images provided by the USGS (http://earthexplorer.usgs.gov) (accessed on 5 December 2021). The resolution of the DEM and RS images was 30 m, the scale of the geological map was 1:50,000, and the scale of the landslide hazard map was 1:10,000. The minimum resolution of the above data was 30 m.

The rainfall data had a temporal resolution and no spatial resolution; therefore, it was necessary to calculate the average annual rainfall and obtain a dataset with a 30 m spatial resolution through resampling. In the selection of the RS image time range, the impact of seasonal characteristics on LU and the ability to distinguish woodland and bare land were considered [55]. RS image data that were not affected by rain, snow or clouds in the TGRA from 2015 to 2020 were selected (as shown in Table 1).

When selecting the RS images from 2017 and 2018, it was found that there were many cumulus clouds in the study area, especially in April, making these images unsuitable for classification. There were no RS images that met the requirements from March to August 2018. Although the cloud cover in March 2018 was as high as 38.22%, it was mainly concentrated in areas outside the study area and had little impact on the study

area; therefore, images of scenes in March 2017 and 2018 were selected. In the other years, images from a scene in April were selected.

**Table 1.** RS image information from 2015 to 2020.

| Landsat 8 | Path/Row | Date Acquired | Overall Cloud Cover (%) | Landsat 8 RS Image in Study Area |
|-----------|----------|---------------|-------------------------|----------------------------------|
| 2015 | | 14 April 2015 | 0.03 | |
| 2016 | | 16 April 2016 | 11.60 | |
| 2017 | | 2 March 2017 | 9.34 | |
| | 125/39 | | | |
| 2018 | | 21 March 2018 | 38.22 | |
| 2019 | | 25 April 2019 | 3.76 | |
| 2020 | | 27 April 2020 | 0.25 | |

### 2.3. Software

The software used in this paper included ENVI 5.3 (https://envi.geoscene.cn/) (accessed on 8 December 2021) for image processing, cropping, radiometric calibration, atmospheric correction and image classification. ArcGIS 10.8 (https://www.esri.com/) (accessed on 10 December 2021) was used for LSM. IBM SPSS Statistics 26 and IBM SPSS Modeler 18 (https://www.ibm.com/) (accessed on 11 December 2021) were used for ANN and SVM modeling and data analysis, and PyTorch 1.7.1 (https://pytorch.org/) (accessed on 12 December 2021) was selected for CNN modeling.

## 3. Methods

The LSM flow chart is shown in Figure 3.

### 3.1. Factor Selection

3.1.1. Factor Correlation and Multicollinearity Analysis

To ensure the independence of each factor and the high accuracy and reliability of the model, it was necessary to test the correlation of the LSM factors by using the Pearson correlation coefficient (PCC). Generally, when the PCC is less than 0.5, the correlation between factors is small. The variance inflation factor (VIF) and tolerance (TOL) were used to test whether there was multicollinearity between variables [56]. When VIF < 5 and TOL > 0.2, there is no multicollinearity problem among independent variables, and they can be used for LSM [57].

### 3.1.2. Relief-F Analysis

The main principle of Relief-F is random feature selection for the parameters that cause landslides, and the weight value of each factor is calculated. The greater the weight assigned to the factor is, the stronger the spatial prediction ability for landslide susceptibility types is, and vice versa. A factor should be removed when the corresponding weight is 0 or a negative value is present, indicating no predictive ability [58,59].

### 3.1.3. PCA

In PCA, uncorrelated output bands are generated from highly correlated multiband data by rotating the coordinate axis to isolate noise and reduce the dimension of the dataset. The principal component bands with large eigenvalues contain more data information and less noise than other bands, and those with smaller eigenvalues contain less information and more noise [60,61]. The first principal component is usually associated with the largest eigenvalues and the largest percentage of data variance, and as the dimension increases, the image quality gradually decreases. Therefore, the first principal component of the NDVI/NDWI/NDBI band is output as the band factor in this paper.

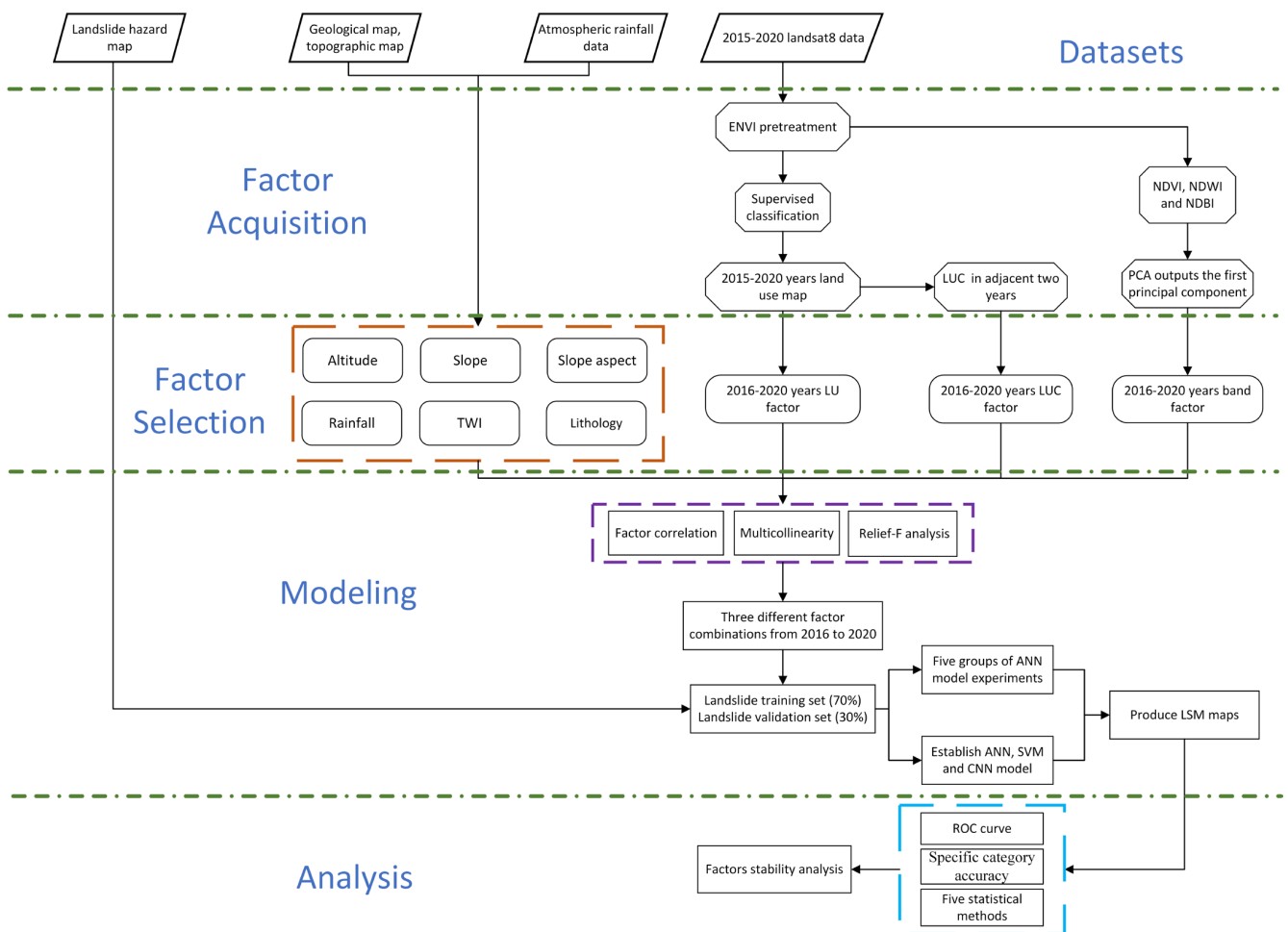

**Figure 3.** Flow chart of LSM.

### 3.2. LSM Model

### 3.2.1. ANN

An ANN is a type of machine learning technique that can be used to complete specific tasks by simulating human thinking and has the ability to learn and generalize from experience [62]. ANNs generally consist of three layers: an input layer (LSM factor), a hidden layer and an output layer (LSM) [63]. "Self-learning" is achieved through forward

propagation and back propagation. If the output result is quite different from the target value, the weight value obtained by multiple cyclic training can be used to minimize the loss function, establish a network with minimized loss and obtain an output value as close as possible to the target value [64]. In this paper, a multilayer perceptron (MLP) ANN is used for LSM. A schematic diagram of the ANN architecture is shown in Figure 4.

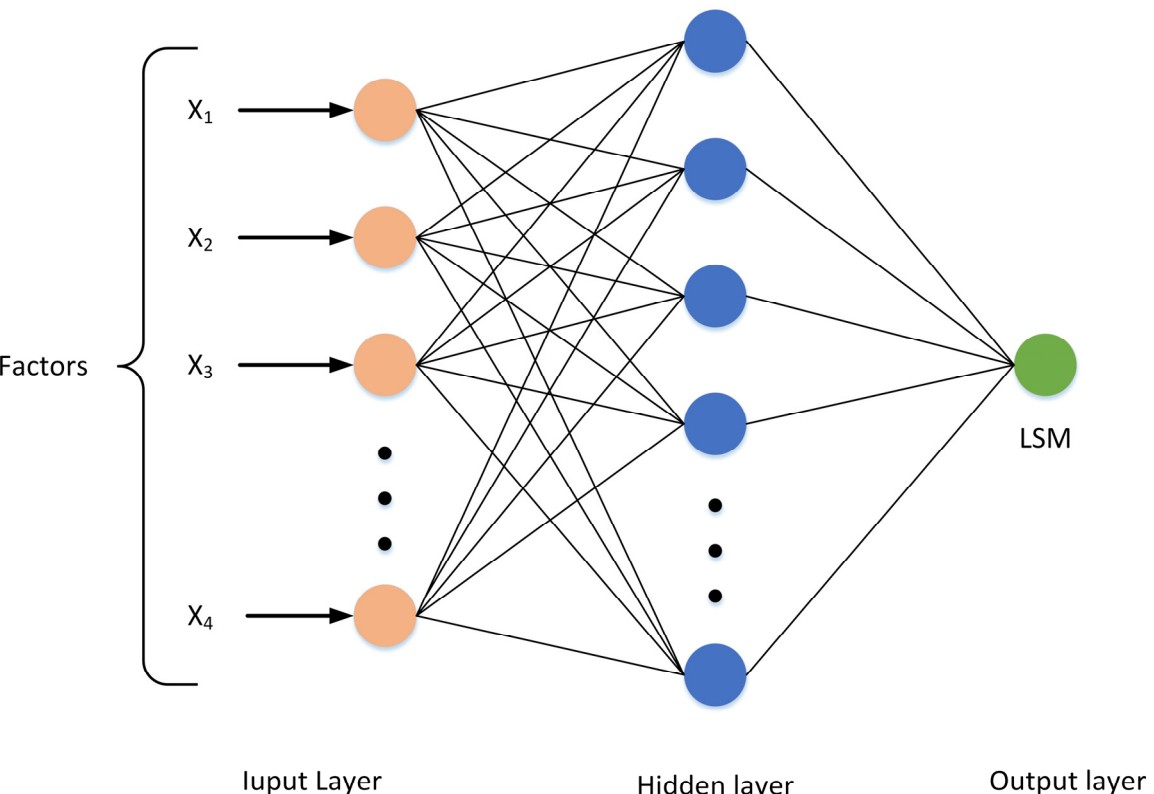

**Figure 4.** ANN architecture diagram.

### 3.2.2. SVM

The first SVM was proposed by Cortes and Vapnik in 1995 [65]. In an SVM, the binary classification problem is initially solved based on a linear discriminant function, and the problem becomes linearly separable by mapping the sample points that are linearly inseparable or difficult to separate in a low-dimensional space to a high-dimensional space [66]. SVMs have many advantages, such as nonlinearity and requiring a small number of samples. Considering the small number of landslides in the study area, an SVM model was used to conduct a sensitivity analysis of landslides [67]. The SVM formula is:

$$\begin{cases} \text{Min} \dfrac{\|w\|^2}{2} \\ st.((w^T x_i) + b)y_i \geq 1 \end{cases} \tag{1}$$

where $x_i$ is a point on the hyperplane, $y_i$ is the classification label set, $i = 1, 2, \ldots, n$ is the number of samples, $w$ is the weight vector related to the direction of the hyperplane, $b$ is the deviation, and $||w||$ is the 2-norm of $w$. In the relaxation of a hard-margin SVM, the values at some points may not be greater than or equal to 1. Thus, Equation (1) can be rewritten as Equation (2):

$$\begin{cases} Min \dfrac{\|w\|^2}{2} + C \sum\limits_{i=1}^{n} \varepsilon_i \\ s.t., ((w^T x_i) + b) \cdot y_1 \geq 1 - \varepsilon_i, \varepsilon_i > 0 \end{cases} \tag{2}$$

where $\varepsilon_i$ is a positive slack variable and $C$ is the penalty factor. A schematic diagram of the SVM architecture is shown in Figure 5. The kernel is a radial basis function (RBF), and the $\gamma$ coefficient of this function is 0.1.

**Figure 5.** SVM architecture diagram.

### 3.2.3. CNN

CNNs were derived from the deep machine learning method used for ANNs. A CNN adopts local connection and weight sharing mechanisms, which not only greatly reduce the number of network parameters but also enhance the generalization effect of the model [68]. In this paper, a one-dimensional CNN is used to evaluate landslide susceptibility, and the landslide data are converted into a one-dimensional vector representation. The basic structure of the CNN includes a convolution layer, an activation function, a pooling layer, a fully connected layer and an output layer. Several feature planes are included in a convolutional layer of the CNN, and each feature plane is composed of neurons arranged in a rectangle. Neurons in the same feature plane share weights (convolution kernels), and the convolution kernels obtain reasonable weights through machine learning. This approach not only reduces the connections between layers of the network but also reduces the risk of overfitting. Pooling can also be seen as a special convolution and subsampling process to simplify model complexity and model parameters [69]. The CNN architecture is shown in Figure 6, and the one-dimensional CNN hyperparameters are shown in Table 2.

**Table 2.** One-dimensional CNN hyperparameters.

| One-Dimensional CNN Hyperparameters | Parameter Values |
| --- | --- |
| Convolution kernel size | $1 \times 4$ |
| Maximum pooling layer kernel size | $1 \times 2$ |
| Activation function | Rectified linear unit |
| Optimizer | Adam |
| Learning rate | 0.01 |
| Batch data size | 2000 |
| Training times | 20 |

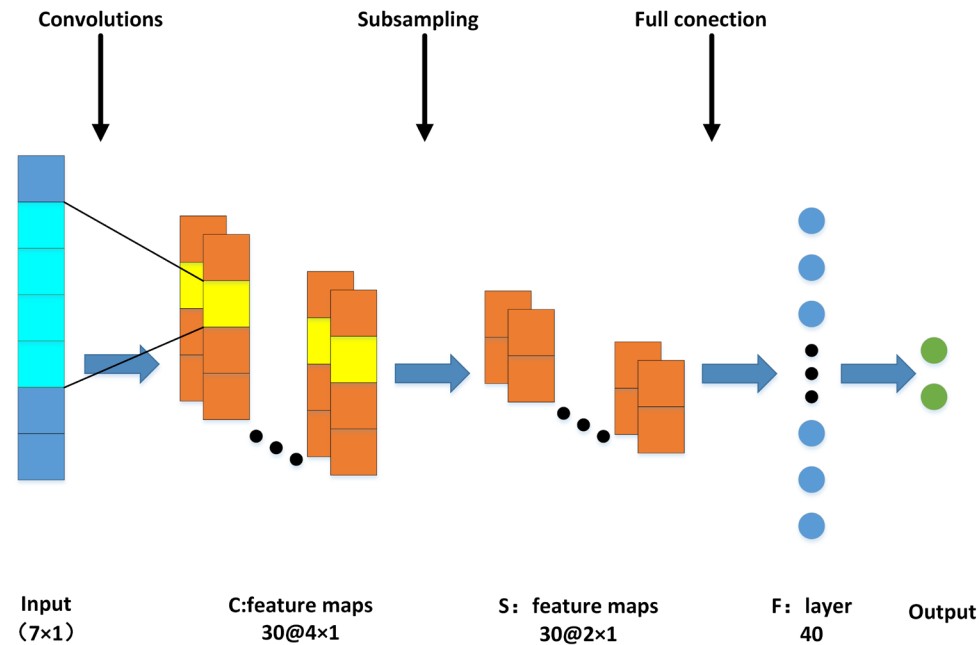

**Figure 6.** CNN architecture diagram.

*3.3. Evaluation Methods*

3.3.1. ROC Curve

The ROC curve originated in the military field, and each point on the curve is related to the same signal stimulus. This method is also often used in LSM [70,71]. The horizontal axis of the ROC curve represents the false-positive rate (FPR), or specificity, and the vertical axis represents the true-positive rate (TPR), or sensitivity. The ROC curve is mainly used in binary classification problems, and the results are divided into positive and negative classes. There are four possible situations: (1) the result is a positive class, and the predicted positive class is the true-positive (TP) class; (2) the result is a positive class, and the prediction is a negative class, that is, a false-negative (FN) class; (3) the result is a negative class, and the prediction is a positive class, that is, a false-positive (FP) class; and (4) the result is a negative class, and the prediction is a negative class, that is, a true-negative (TN) class. These four types of results are used to form the classification matrix of the ROC curve (as shown in Table 3).

**Table 3.** Classification matrix of the ROC curve.

| True | Prediction | |
|---|---|---|
| | **Positive (P)** | **Negative (N)** |
| Positive (P) | True Positive, TP | False Negative, FN |
| Negative (N) | False Positive, FP | True Negative, TN |

The ROC curve can be used to select the best threshold through comparisons of the curves of different machine learning models. In addition, the ROC curves of various models can be plotted in the same coordinate system. The area under the curve (AUC) is enclosed by the ROC curve and the FPR, and it can intuitively reflect the advantages and disadvantages of a given machine learning model.

3.3.2. Specific Category Accuracy

Yu et al. [72,73] proposed a specific category accuracy method to evaluate the accuracy of landslide prediction in different areas, divided the study area into different levels of LSZ, and calculated the prediction accuracy by calculating the proportion of landslide units in

these LSZs. This method is widely used in multiple models. The Equation for calculating the specific category accuracy is:

$$p_i = \frac{A_i}{B_i} \times 100\% \tag{3}$$

where $i = 1, 2, \ldots, S$, $S$ is the number of LSZ categories, $A_i$ is the number of landslides in the $i$-th LSZ category, $B_i$ is the number of the $i$-th LSZ category, and $P_i$ is the specific category accuracy of the $i$-th LSZ category.

### 3.3.3. Statistical Methods

In addition to the above measures used to evaluate model accuracy, five statistical methods were employed to evaluate the model: overall accuracy (OA), precision, recall, the F-measure and the Matthews correlation coefficient (MCC) [74]. OA is the proportion of all correct predictions to the overall sample, and the larger the OA value is, the higher the overall correct prediction rate. Precision is the proportion of correct predictions that are positive compared to the total number of positive predictions, and recall is the proportion of correct predictions that are positive compared to the total number of positive cases. The F-measure is the weighted harmonized average of precision and recall and commonly used to evaluate classification models; when the F-measure is high, it indicates that the test method is effective. The MCC is a correlation coefficient describing the correlation between the actual classification and the predicted classification, with values in the range of −1 to 1; a value of 1 indicates a perfect prediction, a value of 0 indicates that the predicted result is not as good as a randomly predicted result, and a value of −1 indicates that the predicted classification and the actual classification are completely inconsistent.

## 4. Experimental Results and Analysis

### 4.1. Dataset Preparation

(1) According to previous research results regarding LSM factors, six commonly used factors were selected: altitude, slope, slope aspect, rainfall, TWI and lithology [75–79]. These factors are shown in Figure 7.

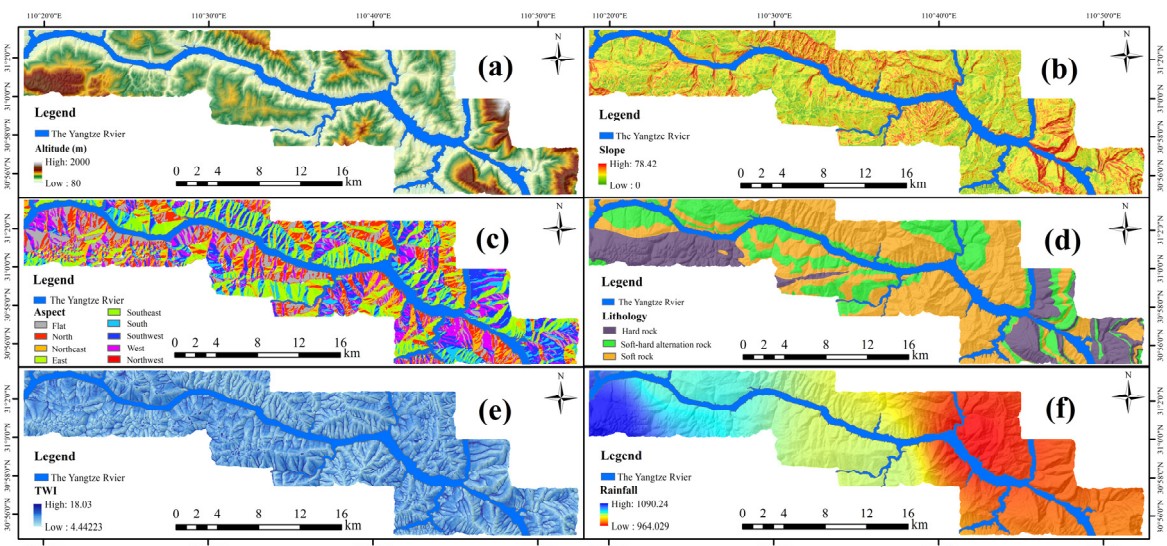

**Figure 7.** Landslide factors in the study area: (**a**) altitude, (**b**) slope, (**c**) slope aspect, (**d**) lithology, (**e**) TWI and (**f**) rainfall.

(2) The selected Landsat 8 RS images were preprocessed based on cropping, radiometric calibration and atmospheric correction procedures with ENVI 5.3 to obtain RS image data in the study area for this experiment. The data were then used to calculate LU and

LUC factors. The SVM model was used to supervise the classification of ground objects in the study area from 2015–2020. The objects were divided into four categories, namely, water bodies, construction land, forest land and bare land (as shown in Figure 8), and the overall accuracy (OA), user accuracy (UA), producer accuracy (PA) and kappa coefficient were used to measure the classification accuracy of LU factor maps (as shown in Table 4).

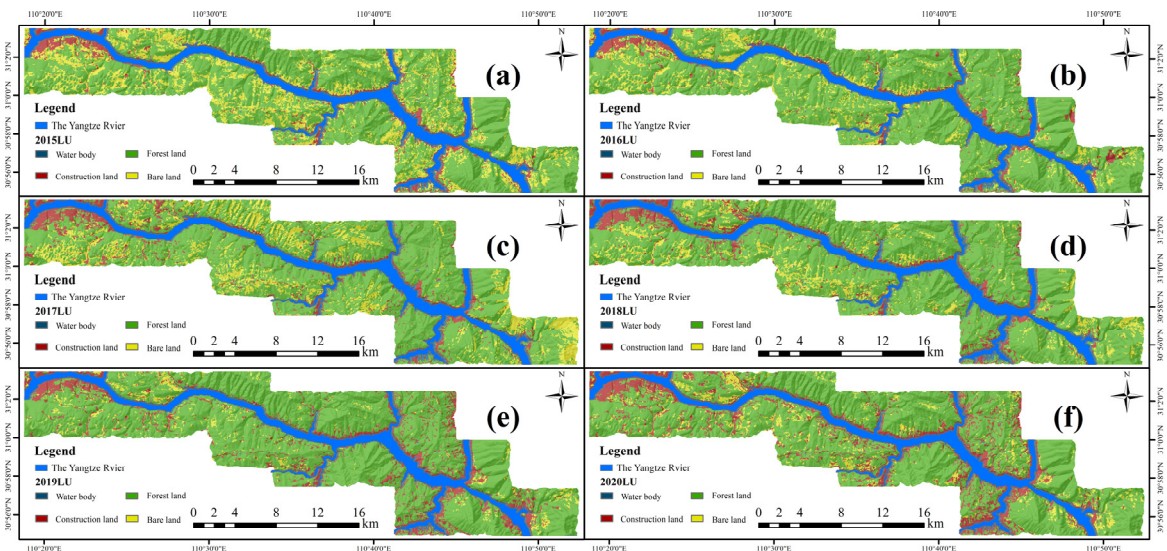

**Figure 8.** (**a**–**f**) are the 2015–2020 LU factors in the study area.

**Table 4.** Classification accuracy index of LU factors.

| Year | LU Factor | PA (%) | UA (%) | OA (%) | Kappa (%) |
|---|---|---|---|---|---|
| 2015 | Water body | 99.71 | 99.07 | 99.19 | 98.76 |
|  | Construction land | 96.20 | 99.07 |  |  |
|  | Forest land | 99.97 | 99.66 |  |  |
|  | Bare land | 96.93 | 96.93 |  |  |
| 2016 | Water body | 97.95 | 99.24 | 96.79 | 95.04 |
|  | Construction land | 96.60 | 92.18 |  |  |
|  | Forest land | 98.98 | 96.29 |  |  |
|  | Bare land | 76.20 | 96.59 |  |  |
| 2017 | Water body | 100 | 98.55 | 98.07 | 97.28 |
|  | Construction land | 97.47 | 99.00 |  |  |
|  | Forest land | 97.92 | 97.58 |  |  |
|  | Bare land | 95.71 | 94.87 |  |  |
| 2018 | Water body | 99.88 | 99.76 | 96.46 | 94.09 |
|  | Construction land | 94.61 | 97.90 |  |  |
|  | Forest land | 97.62 | 96.67 |  |  |
|  | Bare land | 83.16 | 85.56 |  |  |
| 2019 | Water body | 99.62 | 98.69 | 98.64 | 97.90 |
|  | Construction land | 96.56 | 98.76 |  |  |
|  | Forest land | 99.90 | 99.04 |  |  |
|  | Bare land | 90.61 | 93.26 |  |  |
| 2020 | Water body | 99.75 | 98.15 | 97.87 | 96.60 |
|  | Construction land | 96.82 | 92.93 |  |  |
|  | Forest land | 99.71 | 99.76 |  |  |
|  | Bare land | 71.67 | 95.76 |  |  |

The classification results showed that the kappa coefficient was greater than 90% from 2015–2020, indicating that the classification accuracy was good, with limited land type misclassification.

(3) LU and LUC factors were extracted from the LU factor maps (as shown in Figure 9).

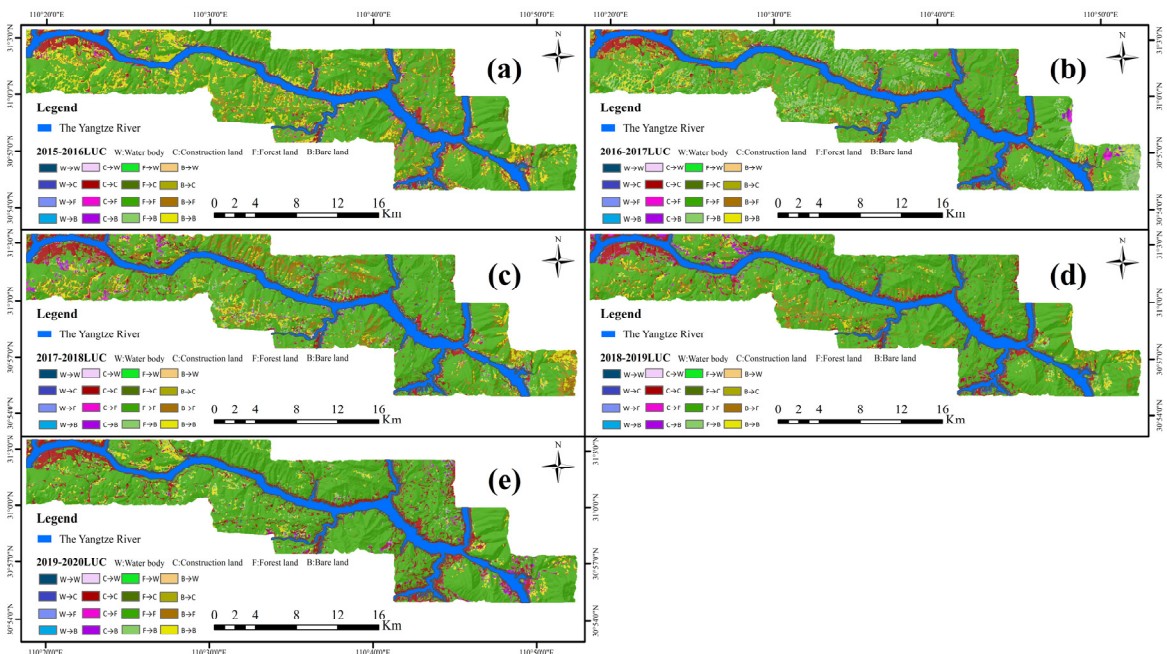

**Figure 9.** (**a**–**e**) are the 2016–2020 LUC factors in the study area. W, C, F and B in the figure denote water bodies, construction land, forest land and bare land.

(4) The formulas for the NDVI, NDWI and NDBI bands are shown in Table 5. From these formulas, the results of the three indices from 2016–2020 were obtained, and the first principal component with the largest amount of information was output with the PCA method to obtain the band factor. The band factors are shown in Figure 10.

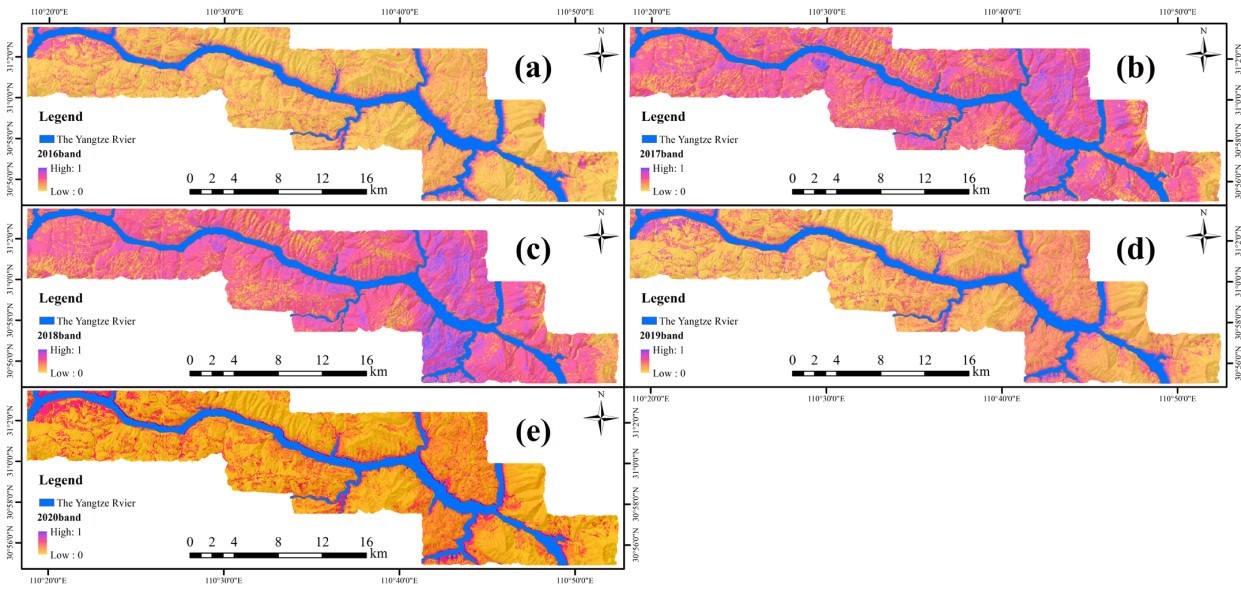

**Figure 10.** (**a**–**e**) are the 2016–2020 band factors in the study area.

**Table 5.** Band math.

| Name | The Formula | Instruction |
|---|---|---|
| NDVI | (band5 − band4)/(band5 + band4) | band 3 is the green band, |
| NDWI | (band3 − band5)/(band3 + band5) | band 4 is the red band, |
| NDBI | (band6 − band5)/(band6 + band5) | band 5 is the infrared, band 6 is the mid-infrared band with a central wavelength of 1.61 μm |

The value ranges of the factors in the study area are shown in Table 6.

**Table 6.** The ranges of landslide factors.

| Category | Subcategory | Factor | Unit | Range |
|---|---|---|---|---|
| Controlling factors | Topography | Altitude | m | 80.00–2000.00 |
| | | Slope | - | 0.00–78.42 |
| | | Aspect | - | (1) Flat, (2) North, (3) Northeast, (4) East, (5) Southeast, (6) South, (7) Southwest, (8) West, (9) Northwest |
| | Geology | Lithology | - | (1) Hard rock, (2) Soft-hard alternation rock, (3) Soft rock |
| | Hydrological | Topographic Wetness Index | - | 4.44–18.03 |
| | Atmospheric precipitation | Rainfall | mm | 964.019–1090.24 |
| Influencing factor | Human engineering activities | LU | - | (1) Water body (W) (2) Construction land (C) (3) Forest land (F) (4) Bare land (B) |
| | | LUC | - | W→ * W, W→C, W→F, W→B, C→W, C→C, C→F, C→B, F→W, F→C, F→F, F→W, B→W, B→C, B→F, B→B |
| | Index | PCA outputs the first principal component of NDVI, NDWI and NDBI | - | 0.00–1.00 |

*: A→B means the object category changes from A to B.

### 4.2. Training and Validation Sets

There are five LSM calculation units: a slope unit, a grid unit, an area unit, a subwatershed unit and an unique condition unit [80]. Grid units are most commonly used in LSM, and their advantage is that their pixels are used as calculation units to ensure that the area of units is the same; thus, they are suitable for LSM models with large amounts of data [81]. Therefore, the grid unit was selected as the LSM calculation unit in this paper.

A buffer zone of 3 raster distances (90 m) at the boundary of the landslide surface was used to eliminate the effect of inaccurate landslide surfaces on the LSM. Landslide locations were randomly selected in a 70/30 ratio for training and validation of the models [82–84]. There were 425,258 computing units in the study area, which constituted the entire sample set. There were a total of 202 landslide surfaces in the study area (25,884 calculation units), and 70% of the landslide surfaces (141 and 17,432 calculation units) were randomly selected. The landslide distribution data were labeled as 1. The remaining 30% of landslide surfaces (61 and 8452 calculation units) and all nonlandslide calculation units in the study area were labeled as 0. Landslide calculation units and the nonlandslide calculation units randomly selected at a ratio of 1:1 were combined into a training set with a total of 34,864 calculation units, and the remaining 30% of the landslide surfaces, encompassing 8452 calculation units, were included in the validation set. The distributions of the training and validation sets in the study area are shown in Figure 11.

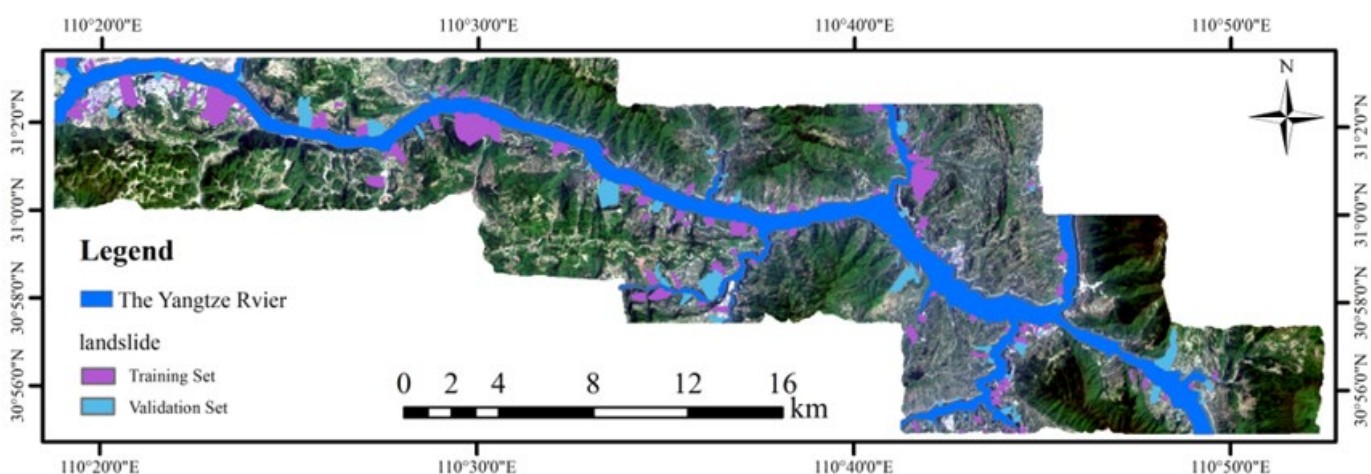

**Figure 11.** The distributions of the validation and training sets in the study area.

*4.3. Experiment*

4.3.1. Factor Analysis and Data Preparation

(1) Analysis of factor correlation and multicollinearity results

All the above factors were divided into three groups: six commonly used factors and LU (named land use factor combination, LUFC), six commonly used factors and LUC (named land use change factor combination, LUCFC), and six commonly used factors and band (named band math factor combination, BMFC). These groups were imported into SPSS Statistics 26 for PCC evaluation and to calculate the VIF for multicollinearity assessment. With the 2016 data as an example, the heatmap of the PCC is shown in Figure 12, and the VIF and TOL values are shown in Table 7.

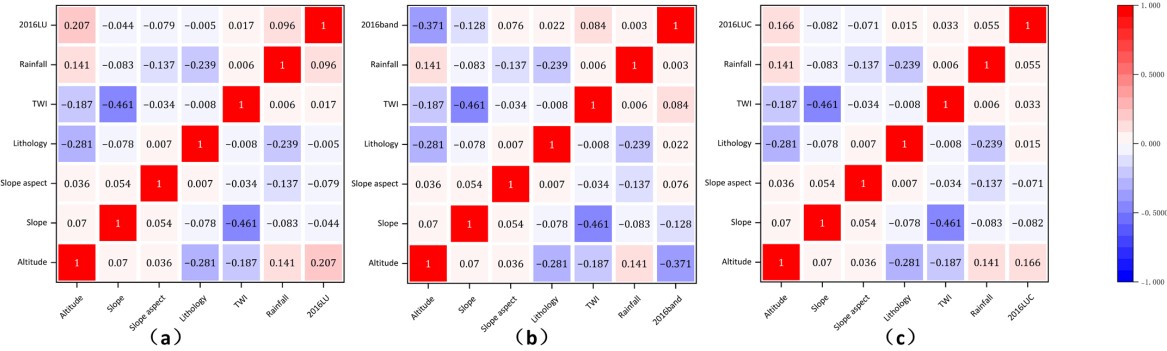

**Figure 12.** (**a**) PCC matrix for the LU factor for 2016, (**b**) PCC matrix for the band factor for 2016, (**c**) PCC matrix for the LUC factor for 2016.

**Table 7.** TOL and VIF for the three factor combinations for 2016.

| Factor | LUFC | | LUCFC | | BMFC | |
|---|---|---|---|---|---|---|
| | TOL | VIF | TOL | VIF | TOL | VIF |
| Altitude | 0.835 | 1.198 | 0.846 | 1.182 | 0.746 | 1.341 |
| Lithology | 0.768 | 1.302 | 0.765 | 1.307 | 0.756 | 1.323 |
| Slope | 0.971 | 1.030 | 0.972 | 1.029 | 0.965 | 1.037 |
| Slope aspect | 0.860 | 1.163 | 0.860 | 1.163 | 0.856 | 1.168 |
| TWI | 0.753 | 1.328 | 0.753 | 1.327 | 0.752 | 1.329 |
| Rainfall | 0.903 | 1.108 | 0.906 | 1.104 | 0.905 | 1.105 |
| LU/LUC/band | 0.938 | 1.066 | 0.953 | 1.049 | 0.830 | 1.205 |

Figure 12 shows that the correlation between the factors was low, among which the negative correlation between the TWI and slope was the largest ($-0.461$), but it was still less than $-0.5$, indicating that the correlation between these two factors was weak and had no adverse effect on the establishment of the LSM model. Table 5 shows that TOL was >0.2 and VIF was <5, indicating that there was no multicollinearity problem among the factors.

(2)    Relief-F analysis

With the 2016 data as an example, the LUFC, LUCFC and BMFC factor combinations were input into the Relief-F algorithm for analysis. The Relief-F results are shown in Figure 13.

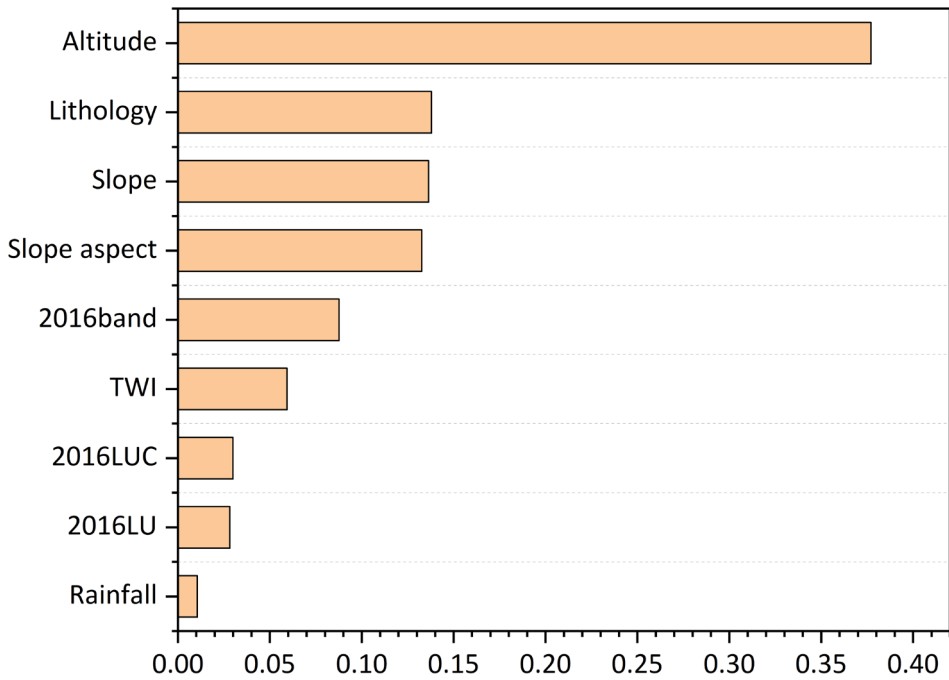

**Figure 13.** Relief-F results for three factor combinations for 2016.

After calculations, it was found that in the Relief-F algorithm, all factor weights were greater than zero and could be used for LSM.

(3)    Data preparation

Before inputting the factors into the model for modeling, it was necessary to normalize continuous factors such as altitude, slope, rainfall and TWI. The purpose was to eliminate the influence of different factor units on the LSM results, improve the speed of the gradient descent method to find the optimal solution in the CNN model, and improve the accuracy of the model.

4.3.2. Experimental Results

LUFC, LUCFC and BMFC were input into the ANN model to obtain the respective landslide susceptibility index (LSI) values. The LSI is a continuous variable that ranges from 0–1; the closer to 1 the LSI value is, the greater the probability of landslides, and vice versa. The LSIs of the three factor combinations based on the ANN model are shown in Figure 14.

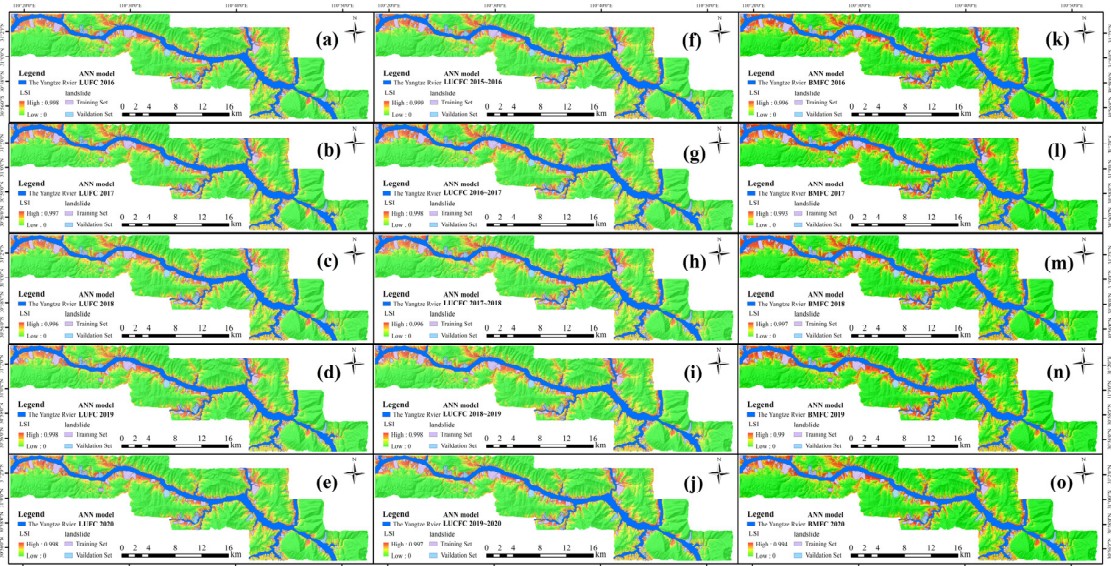

**Figure 14.** LSIs of the three factor combinations based on the ANN model: (**a**–**e**) show the LSM of the ANN-based LUFC from 2016 to 2020; (**f**–**j**) show the LSM of the ANN-based LUCFC from 2016 to 2020; and (**k**–**o**) show the LSM of the ANN-based BMFC from 2016 to 2020.

To intuitively reflect the results of the LSI and increase comprehension, the LSI was divided into five categories to generate LSZs, with values of 0–0.5, 0.5–0.75, 0.75–0.85, 0.85–0.95 and 0.95–1 denoting very low susceptibility areas, low susceptibility areas, moderate susceptibility areas, high susceptibility areas and very high susceptibility areas, respectively [72]. The LSZs of the three factor combinations were obtained, as shown in Figure 15.

### 4.3.3. Evaluation of the Experimental Results

The ROC curves and AUC values of the ANN model based on the maps of the three factor combinations are shown in Figure 16 and Table 8. The larger the AUC value was (that is, the larger the area enclosed by the curve and the X-axis), the higher the accuracy of the model [85–87].

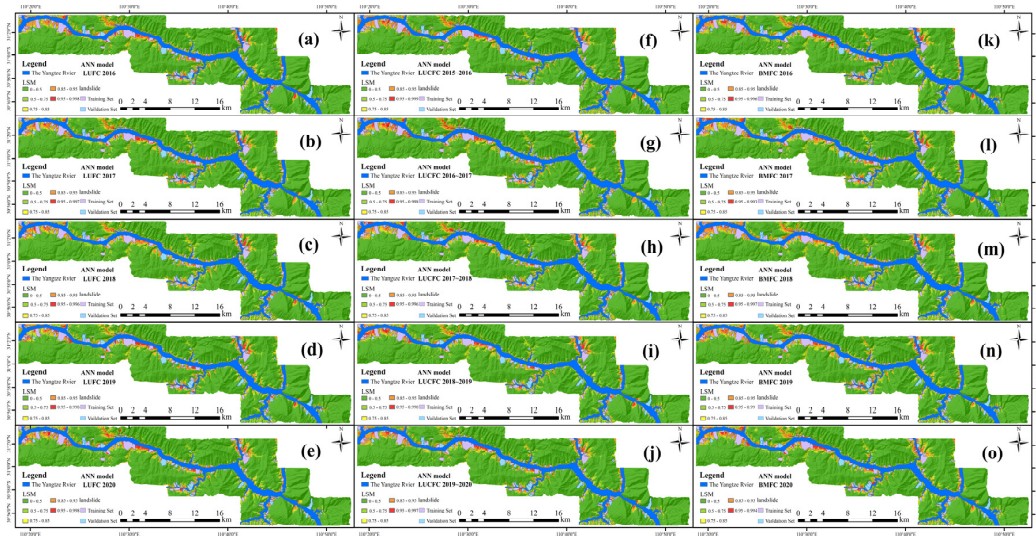

**Figure 15.** LSZs of the three factor combinations based on the ANN model: (**a**–**e**) show the LSM of the ANN-based LUFC from 2016 to 2020; (**f**–**j**) show the LSM of the ANN-based LUCFC from 2016 to 2020; and (**k**–**o**) show the LSM of the ANN-based BMFC from 2016 to 2020.

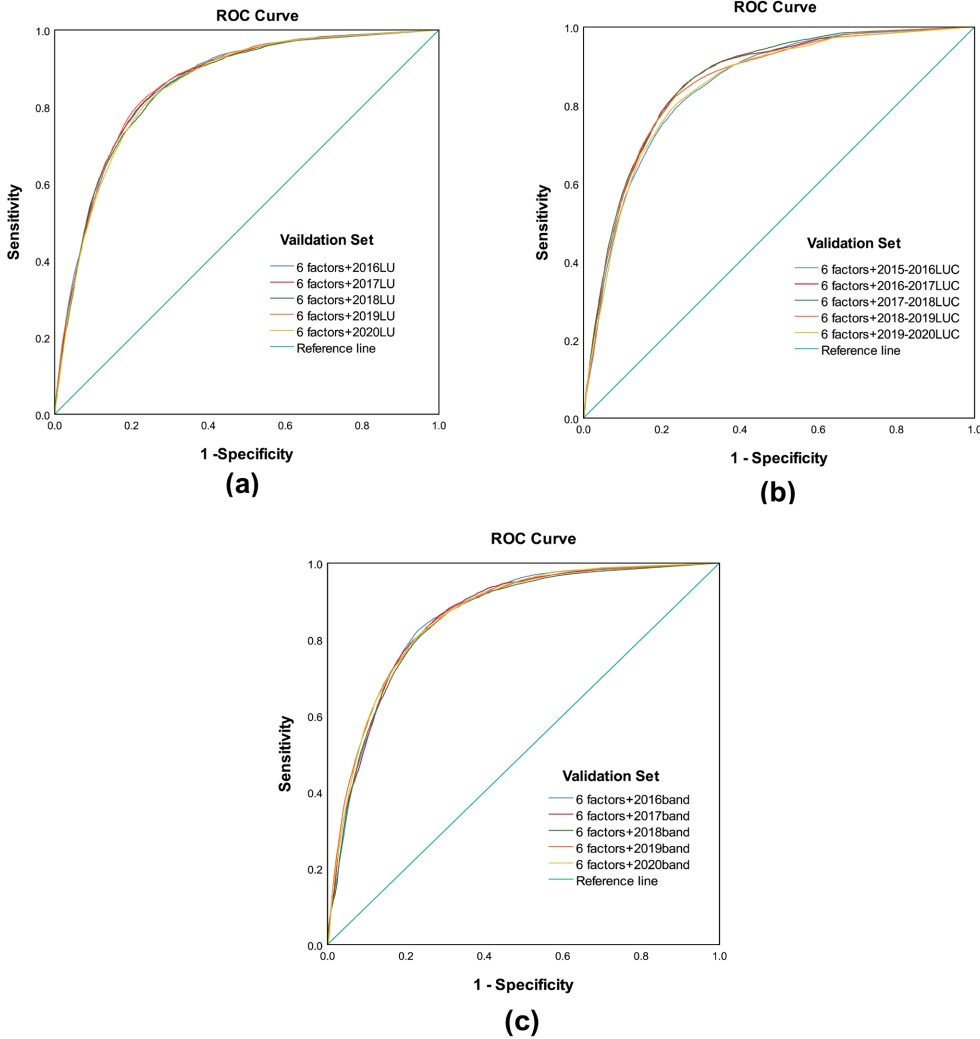

**Figure 16.** ROC curve analysis of the three factor combinations based on the ANN model: (**a**) ROC curve for LUFC; (**b**) ROC curve for LUCFC; (**c**) ROC curve for BMFC.

**Table 8.** AUC values of the ANN model based on the three factor combinations.

| Year | Area Test Result Variable (S) | Area | Asymptotic 95% Confidence Interval | |
|---|---|---|---|---|
| | | | Lower Bound | Upper Bound |
| 2016 | LUFC | 0.856 | 0.852 | 0.860 |
| | LUCFC | 0.849 | 0.845 | 0.852 |
| | BMFC | 0.860 | 0.857 | 0.864 |
| 2017 | LUFC | 0.857 | 0.853 | 0.860 |
| | LUCFC | 0.859 | 0.855 | 0.862 |
| | BMFC | 0.859 | 0.856 | 0.863 |
| 2018 | LUFC | 0.853 | 0.849 | 0.856 |
| | LUCFC | 0.864 | 0.860 | 0.867 |
| | BMFC | 0.854 | 0.850 | 0.857 |
| 2019 | LUFC | 0.858 | 0.854 | 0.862 |
| | LUCFC | 0.855 | 0.852 | 0.859 |
| | BMFC | 0.863 | 0.859 | 0.866 |
| 2020 | LUFC | 0.851 | 0.848 | 0.855 |
| | LUCFC | 0.847 | 0.843 | 0.851 |
| | BMFC | 0.863 | 0.859 | 0.866 |

Figure 16 and Table 8 show that the highest AUC value was observed for LUCFC in 2018 (0.864), and the highest AUC values in other years were as follows: 0.860 for BMFC in 2016, 0.859 for BMFC and LUCFC in 2017, 0.863 for BMFC in 2019 and 0.863 for BMFC in 2020.

The specific category accuracies of the very high susceptibility areas identified with the ANN models based on the three factor combinations are shown in Table 9.

**Table 9.** The specific category accuracies of the very high susceptibility areas identified with the ANN models based on the three factor combinations.

| Year | Factor | Category of Susceptibility | | | | |
|------|--------|----------|-----|--------|------|-----------|
| | | **Very Low** | **Low** | **Medium** | **High** | **Very High** |
| 2016 | LUFC (%) | 0.74 | 5.54 | 7.71 | 11.42 | 17.01 |
| | LUCFC (%) | 0.82 | 5.07 | 8.58 | 11.71 | 13.31 |
| | BMFC (%) | 0.72 | 5.76 | 8.09 | 11.85 | 14.92 |
| 2017 | LUFC (%) | 0.72 | 5.31 | 8.97 | 11.13 | 15.81 |
| | LUCFC (%) | 0.77 | 5.37 | 8.07 | 11.98 | 15.60 |
| | BMFC (%) | 0.68 | 5.66 | 6.41 | 12.28 | 15.87 |
| 2018 | LUFC (%) | 0.73 | 4.77 | 10.11 | 11.41 | 14.36 |
| | LUCFC (%) | 0.74 | 5.19 | 9.08 | 12.60 | 15.44 |
| | BMFC (%) | 0.73 | 4.78 | 7.68 | 11.51 | 17.53 |
| 2019 | LUFC (%) | 0.72 | 5.70 | 8.74 | 10.97 | 15.48 |
| | LUCFC (%) | 0.70 | 5.52 | 8.68 | 12.80 | 13.28 |
| | BMFC (%) | 0.69 | 4.54 | 6.78 | 12.49 | 18.84 |
| 2020 | LUFC (%) | 0.79 | 5.42 | 8.44 | 11.79 | 13.27 |
| | LUCFC (%) | 0.79 | 5.72 | 9.61 | 10.32 | 15.47 |
| | BMFC (%) | 0.70 | 4.44 | 8.32 | 11.98 | 17.75 |

Table 9 shows that the highest specific category accuracy of very high susceptibility areas (18.84%) was obtained for BMFC in 2019. In other years, the highest values (15.87% in 2017, 17.53% in 2018 and 17.75% in 2020) were also associated with BMFC, except in 2016, when the highest value corresponded to LUFC (17.01%).

The results of the five statistical methods based on ANN models and the three factor combinations are shown in Table 10.

**Table 10.** The results of the five statistical methods based on ANN models for the three factor combinations.

| Year | Factor | Five Statistical Methods | | | | |
|------|--------|--------|-----------|--------|-----------|------|
| | | **OA (%)** | **Precision** | **Recall** | **F-Measure** | **MCC** |
| 2016 | LUFC | 83.85 | 0.0859 | 0.7041 | 0.1531 | 0.2425 |
| | LUCFC | 84.18 | 0.0859 | 0.6742 | 0.1523 | 0.2398 |
| | BMFC | 83.60 | 0.0857 | 0.7146 | 0.1530 | 0.2432 |
| 2017 | LUFC | 83.40 | 0.0847 | 0.7147 | 0.1515 | 0.2418 |
| | LUCFC | 84.48 | 0.0853 | 0.6823 | 0.1517 | 0.2398 |
| | BMFC | 82.51 | 0.0824 | 0.7338 | 0.1482 | 0.2398 |
| 2018 | LUFC | 82.77 | 0.0816 | 0.7127 | 0.1465 | 0.2370 |
| | LUCFC | 84.08 | 0.0870 | 0.7034 | 0.1549 | 0.2441 |
| | BMFC | 82.65 | 0.0813 | 0.7149 | 0.1459 | 0.2366 |
| 2019 | LUFC | 83.60 | 0.0854 | 0.7118 | 0.1526 | 0.2426 |
| | LUCFC | 83.35 | 0.0852 | 0.7221 | 0.1524 | 0.2431 |
| | BMFC | 82.81 | 0.0831 | 0.7263 | 0.1491 | 0.2402 |

**Table 10.** *Cont*.

| Year | Factor | Five Statistical Methods | | | | |
|------|--------|--------|-----------|--------|-----------|--------|
| | | OA (%) | Precision | Recall | F-Measure | MCC |
| | LUFC | 83.76 | 0.0836 | 0.6859 | 0.1491 | 0.2377 |
| 2020 | LUCFC | 84.19 | 0.0856 | 0.6842 | 0.1522 | 0.2403 |
| | BMFC | 82.97 | 0.0834 | 0.7220 | 0.1495 | 0.2404 |

Table 10 shows that the highest OA value was obtained for LUCFC in 2017 (84.48%), the highest precision was obtained for LUCFC in 2018 (0.0870), the highest recall was obtained for BMFC in 2017 (0.7338), the highest F-measure value was obtained for LUCFC in 2018 (0.1549) and the highest MCC was obtained for LUCFC in 2018 (0.2441).

Tables 8–10 show that BMFC was associated with the highest AUC values in all years except 2018, and BMFC displayed the highest specific category accuracy values of very high susceptibility areas in all years except 2016. In general, the BMFC results obtained with the ANN model were better than those for LUFC and LUCFC, indicating that the band factor was more important in the ANN model than the LUC and LU factors.

## 5. Discussion

### 5.1. Analysis of ANN, SVM and CNN Results

To verify the extent to which the band factor plays an important role in LSM, two other commonly used models, an SVM and a CNN, were selected for comparison. The LSM results of these two models were analyzed using the ROC curve, the specific category accuracy of very high susceptibility areas and five statistical methods.

The AUC values of the three models based on the three factor combinations are shown in Table 11.

**Table 11.** The AUC values of the three models based on the three factor combinations.

| Year | Area Test Result Variable (S) | AUC | | |
|------|-------------------------------|-----|-----|-----|
| | | ANN | SVM | CNN |
| | LUFC | 0.856 | 0.848 | 0.832 |
| 2016 | LUCFC | 0.849 | 0.847 | 0.821 |
| | BMFC | 0.860 | 0.849 | 0.832 |
| | LUFC | 0.857 | 0.853 | 0.806 |
| 2017 | LUCFC | 0.859 | 0.851 | 0.798 |
| | BMFC | 0.859 | 0.851 | 0.823 |
| | LUFC | 0.853 | 0.847 | 0.818 |
| 2018 | LUCFC | 0.864 | 0.850 | 0.780 |
| | BMFC | 0.854 | 0.854 | 0.832 |
| | LUFC | 0.858 | 0.849 | 0.828 |
| 2019 | LUCFC | 0.855 | 0.846 | 0.770 |
| | BMFC | 0.863 | 0.850 | 0.829 |
| | LUFC | 0.851 | 0.846 | 0.818 |
| 2020 | LUCFC | 0.847 | 0.846 | 0.787 |
| | BMFC | 0.863 | 0.849 | 0.833 |

Table 11 shows that for the SVM model, the highest AUC values from 2016–2020 (except in 2017) were all obtained for BMFC; among them, the highest AUC value was obtained for BMFC in 2018 (0.854). For the CNN model, the highest AUC values from 2016–2020 were all associated with BMFC, and the highest AUC value was obtained for BMFC in 2020 (0.833).

The specific category accuracies of very high susceptibility areas for the three models based on the three factor combinations are shown in Table 12.

**Table 12.** The specific category accuracies of very high susceptibility areas for the three models based on the three factor combinations.

| Year | Factor | Very High Category of Susceptibility | | |
|---|---|---|---|---|
| | | ANN | SVM | CNN |
| 2016 | LUFC (%) | 17.01 | 13.40 | 14.44 |
| | LUCFC (%) | 13.31 | 14.22 | 7.46 |
| | BMFC (%) | 14.92 | 12.84 | 11.82 |
| 2017 | LUFC (%) | 15.81 | 13.61 | 8.33 |
| | LUCFC (%) | 15.60 | 13.88 | 8.57 |
| | BMFC (%) | 15.87 | 13.98 | 15.95 |
| 2018 | LUFC (%) | 14.36 | 15.58 | 7.04 |
| | LUCFC (%) | 15.44 | 14.54 | 7.75 |
| | BMFC (%) | 17.53 | 15.26 | 12.23 |
| 2019 | LUFC (%) | 15.48 | 15.16 | 11.82 |
| | LUCFC (%) | 13.28 | 13.78 | 3.36 |
| | BMFC (%) | 18.84 | 13.48 | 10.61 |
| 2020 | LUFC (%) | 13.27 | 14.82 | 12.49 |
| | LUCFC (%) | 15.47 | 14.60 | 9.07 |
| | BMFC (%) | 17.75 | 13.04 | 12.33 |

Table 12 shows that for the SVM model, the highest specific category accuracy value of very high susceptibility areas was obtained for LUFC in 2018 (15.58%), and the highest values in other years were obtained for LUCFC (14.22%) in 2016, BMFC (13.98%) in 2017, LUFC (15.16%) in 2019 and LUFC (14.82%) in 2020. For the CNN model, the highest specific category accuracy value of very high susceptibility areas was obtained for BMFC in 2017 (15.95%), and the highest values in other years were obtained for LUFC in 2016 (14.44%), BMFC in 2018 (12.23%), LUFC in 2019 (11.82%) and LUFC in 2020 (12.49%).

Statistical analyses of the results of the SVM and CNN models based on the three factor combinations were performed, and the results are shown in Table 13.

**Table 13.** Statistical analysis of the results of the SVM and CNN models based on the three factor combinations.

| Model | Year | Factor | Five Statistical Methods | | | | |
|---|---|---|---|---|---|---|---|
| | | | OA (%) | Precision | Recall | F-Measure | MCC |
| SVM | 2016 | LUFC | 83.25 | 0.0814 | 0.6881 | 0.1456 | 0.2347 |
| | | LUCFC | 83.62 | 0.0829 | 0.6770 | 0.1477 | 0.2359 |
| | | BMFC | 83.52 | 0.0826 | 0.6868 | 0.1474 | 0.2363 |
| | 2017 | LUFC | 83.39 | 0.0816 | 0.6840 | 0.1459 | 0.2347 |
| | | LUCFC | 83.69 | 0.0820 | 0.6746 | 0.1462 | 0.2344 |
| | | BMFC | 83.37 | 0.0841 | 0.7094 | 0.1503 | 0.2404 |
| | 2018 | LUFC | 83.21 | 0.0806 | 0.6823 | 0.1442 | 0.2331 |
| | | LUCFC | 83.65 | 0.0818 | 0.6729 | 0.1458 | 0.2339 |
| | | BMFC | 83.44 | 0.0846 | 0.7113 | 0.1512 | 0.2413 |
| | 2019 | LUFC | 83.32 | 0.0817 | 0.6875 | 0.1460 | 0.2351 |
| | | LUCFC | 83.66 | 0.0827 | 0.6819 | 0.1475 | 0.2361 |
| | | BMFC | 83.45 | 0.0829 | 0.6934 | 0.1481 | 0.2373 |
| | 2020 | LUFC | 83.42 | 0.0819 | 0.6848 | 0.1463 | 0.2351 |
| | | LUCFC | 83.65 | 0.0826 | 0.6808 | 0.1473 | 0.2358 |
| | | BMFC | 83.45 | 0.0825 | 0.6893 | 0.1473 | 0.2363 |

**Table 13.** *Cont.*

| Model | Year | Factor | Five Statistical Methods | | | | |
|-------|------|--------|--------|-----------|--------|-----------|-----|
| | | | **OA (%)** | **Precision** | **Recall** | **F-Measure** | **MCC** |
| CNN | 2016 | LUFC | 82.33 | 0.0736 | 0.6491 | 0.1322 | 0.2203 |
| | | LUCFC | 80.79 | 0.0658 | 0.6503 | 0.1195 | 0.2087 |
| | | BMFC | 75.34 | 0.0609 | 0.7553 | 0.1127 | 0.2052 |
| | 2017 | LUFC | 82.34 | 0.0700 | 0.6113 | 0.1255 | 0.2120 |
| | | LUCFC | 81.64 | 0.0728 | 0.6022 | 0.1298 | 0.2158 |
| | | BMFC | 86.06 | 0.0783 | 0.5309 | 0.1364 | 0.2143 |
| | 2018 | LUFC | 82.82 | 0.0701 | 0.5942 | 0.1255 | 0.2107 |
| | | LUCFC | 80.19 | 0.0623 | 0.6084 | 0.1130 | 0.2008 |
| | | BMFC | 82.79 | 0.0736 | 0.6296 | 0.1318 | 0.2186 |
| | 2019 | LUFC | 79.47 | 0.0661 | 0.6782 | 0.1205 | 0.2110 |
| | | LUCFC | 72.27 | 0.0487 | 0.6677 | 0.0908 | 0.1817 |
| | | BMFC | 79.99 | 0.0699 | 0.7024 | 0.1271 | 0.2183 |
| | 2020 | LUFC | 80.92 | 0.0665 | 0.6285 | 0.1202 | 0.2083 |
| | | LUCFC | 84.95 | 0.0687 | 0.4981 | 0.1207 | 0.1988 |
| | | BMFC | 83.83 | 0.0772 | 0.6210 | 0.1374 | 0.2229 |

Table 13 shows that for the SVM model, the highest OA value was obtained for LUCFC in 2017 (83.69%), the highest precision was obtained for BMFC in 2018 (0.0846), the highest recall was obtained for BMFC in 2018 (0.7113), the highest F-measure value was obtained for BMFC in 2018 (0.1512) and the highest MCC was obtained for BMFC in 2018 (0.2413). For the CNN model, the highest OA value was obtained for BMFC in 2017 (86.06%), the highest precision was obtained for BMFC in 2017 (0.0783), the highest recall was obtained for BMFC in 2016 (0.7553), the highest F-measure was obtained for BMFC in 2020 (0.1374) and the highest MCC was obtained for BMFC in 2020 (0.2229).

For the ANN model, the highest AUC values (except in 2018) were all obtained for BMFC, the highest specific category accuracy values of very high susceptibility areas were all obtained for BMFC (except in 2016), and the OA, precision, F-measure and MCC values of the LUCFC in the analysis of the five statistical methods were the highest among those shown. For the SVM model, the highest AUC values were all obtained for BMFC (except in 2017), and the specific category accuracy values of very high susceptibility areas based on LUFC, LUCFC and BMFC were highest between 2016 and 2020. Among the five statistical methods, all the highest values were associated with the BMFC, except for the highest OA value, which corresponded to the LUCFC. For the CNN model, the highest AUC values between 2016 and 2020 were all obtained for BMFC, the highest specific category accuracy values of very high susceptibility areas in 2017 and 2018 were obtained for BMFC, and the highest five statistical methods were all obtained for BMFC. In general, LUFC, LUCFC and BMFC from 2016–2020 had advantages and disadvantages with respect to the AUC values, specific category accuracy values of very high susceptibility areas and the five statistical methods based on the three models. However, generally, the results for BMFC were better than those for LUFC and LUCFC, indicating that the band factor had a better impact on LSM than did the LU and LUC factors.

*5.2. Simple Quantitative Ranking Analysis*

To more intuitively reflect the importance of the three factors, namely, LU, LUC and band, in LSM, a simple ranking method was used to score different factor combinations. The higher the score was, the better the prediction performance. In this study, the scoring principles were as follows. The scores obtained for the AUC value, specific category accuracy value of very high susceptibility areas and the five statistical methods (OA, precision, recall, F-measure and MCC) for LUFC, LUCFC and BMFC, in order from highest to lowest, were ranked as 3 points, 2 points and 1 point, respectively. The scores of the AUC

value, the specific category accuracy value of very high susceptibility areas and the average of the five statistical methods were added together, with the highest scores indicating the best performance.

The scores of the three factor combinations based on the three models are shown in Table 14.

**Table 14.** Performance comparison of three factor combinations based on three models.

| The Score | ANN | | | SVM | | | CNN | | |
|---|---|---|---|---|---|---|---|---|---|
| | LUFC | LUCFC | BMFC | LUFC | LUCFC | BMFC | LUFC | LUCFC | BMFC |
| AUC | 8 | 9 | 15 | 9 | 6 | 13 | 11 | 5 | 15 |
| Very high | 9 | 7 | 14 | 6 | 11 | 13 | 11 | 7 | 12 |
| Five statistical methods' AVG | 11 | 10.8 | 8.2 | 11.6 | 11.6 | 6.8 | 9.8 | 8 | 12.2 |
| Total | 28 | 26.8 | 37.2 | 26.6 | 28.6 | 32.8 | 31.8 | 20 | 39.2 |

Table 14 shows that for the ANN model, the highest AUC score was obtained for BMFC (15 points), the highest specific category accuracy score of very high susceptibility areas was obtained for BMFC (14 points) and the highest average score of the five statistical methods was obtained for LUFC (11 points). The highest overall score of the three was obtained for BMFC (37.2 points). For the SVM model, the highest AUC score was obtained for BMFC (13 points), the highest score for the specific category accuracy of very high susceptibility areas was obtained for BMFC (13 points) and the highest average scores for the five statistical methods were obtained for LUFC and LUCFC (11.6 points). The highest comprehensive score among the three cases was obtained for BMFC (32.8 points). For the CNN model, the highest AUC value (15 points), the highest specific category accuracy of very high susceptibility areas (12 points), and the highest average value of the five statistical methods (12.2 points) were all obtained for BMFC. Moreover, the highest composite score of the three was obtained for BMFC (39.2 points). In summary, BMFC yielded the highest scores for all three models. The LUFC and LUCFC results varied for the three models, and the values were all lower than those for BMFC, which indicates that the band factor was more important than the LUC and LU factors in the three models.

Notably, the band factor from 2016–2020 was obtained using the PCA algorithm by outputting the first principal component of the NDVI/NDBI/NDW bands, that is, the maximum amount of information. The PCA table for the bands from 2016–2020 is shown in Table 15.

**Table 15.** PCA values for the three models.

| Year | PC | Eigenvalue * | Percentage ** |
|---|---|---|---|
| 2016 | 1st | 0.8108 | 85.95% |
| | 2nd | 0.1229 | 13.03% |
| | 3rd | 0.0097 | 1.02% |
| 2017 | 1st | 1.1808 | 98.81% |
| | 2nd | 0.0142 | 1.19% |
| | 3rd | 0.0000 | 0% |
| 2018 | 1st | 0.6469 | 79.63% |
| | 2nd | 0.1359 | 16.73% |
| | 3rd | 0.0296 | 3.64% |
| 2019 | 1st | 0.8336 | 91.41% |
| | 2nd | 0.0732 | 8.04% |
| | 3rd | 0.0050 | 0.55% |

**Table 15.** *Cont*.

| Year | PC | Eigenvalue * | Percentage ** |
|---|---|---|---|
| | 1st | 0.8336 | 87.68% |
| 2020 | 2nd | 0.1062 | 11.16% |
| | 3rd | 0.0110 | 1.16% |

*: Eigenvalue represents the eigenvalues of the first, second and third principal components. **: Percentage represents the percentage of information contained in the first, second and third principal components.

Table 15 shows that the information content values of the first principal component of the PCA output fusion band from 2016–2020 were 85.95, 98.81, 79.63, 91.41 and 87.68%. The smallest amount of information was conveyed by this component in 2018 (79.63%), and according to the results in Tables 11–13. In some of the evaluation methods used for the ANN model, BMFC yielded better results than LUFC and LUCFC. For the SVM model, the highest AUC and the highest values of precision, recall, the F-measure and the MCC among the five statistical methods were all obtained for BMFC. For the CNN model, the highest AUC, the highest specific category accuracy of very high susceptibility areas, and the highest values of precision, recall, the F-measure and the MCC among the five statistical methods were all obtained for BMFC. In summary, even in 2018, when information provided by the first principal component in PCA was the most limited, the LSM results based on BMFC were the best for the three different models.

A simple ranking of the scores for LUFC, LUCFC and BMFC revealed that the highest scores were obtained for BMFC in all three models, and the highest predictive ability for BMFC was obtained for the CNN model, followed by the ANN model and then the SVM model. Although the CNN model displayed good prediction ability for BMFC, it yielded the worst results for LUCFC with a temporal dimension, and LUCFC did not provide advantages in the three models; notably, the prediction ability for LUCFC was lower than that for LUFC. The reason for this result may have been that the LUC in this study was only obtained by detecting changes in the data at an interval of one year, and there was no shorter time interval. Thus, the temporality of LUC was not reflected well.

*5.3. Stability Analysis*

To confirm the stability of the band factor, four groups of experiments were performed using the ANN model. The ROC curve analysis results in 2016 are used as an example, and the AUC values of the three factor combinations of the five groups of ANNs are shown in Table 16. The stability results are shown in Figure 17.

**Table 16.** AUC values of the five groups of ANNs.

| Factor / AUC | Test Gruops | ANN (A) | ANN (B) | ANN (C) | ANN (D) | ANN (E) |
|---|---|---|---|---|---|---|
| | LUFC | 0.856 | 0.852 | 0.882 | 0.855 | 0.854 |
| | LUCFC | 0.849 | 0.856 | 0.874 | 0.848 | 0.853 |
| | BMFC | 0.860 | 0.860 | 0.860 | 0.860 | 0.860 |

Figure 17 shows that the AUC value for BMFC was lower than those for LUFC and LUCFC only in the third set of ANN experiments. The standard deviations of the three factor combinations were 0.025 for LUFC, 0.022 for LUCFC and 0 for BMFC. In LSM, the stability of BMFC was better than that of LUFC and LUCFC, indicating that the stability of the band factor was better than that of the LU and LUC factors.

Research has shown that LU is only considered a dynamic factor when it changes over decades or even centuries, and it is considered to be a static factor in a short period [88]. Intuitively, the experiments in this paper confirm that there is no significant difference between the results of LUC and LU factors in LSM. Second, due to the variable quality of RS images, the LU data obtained by manual classification and the LUC data obtained by detecting changes in LU data over time are often incorrectly classified. Moreover, the setting

of image data parameters, the selection of image processing software and the limitations of modeling methods all influence the image classification accuracy [89], which in turn affects the results of LSM. Considering these problems, the first principal component (band) of the NDVI/NDWI/NDBI-based results obtained by using the PCA algorithm was added for classification in this study. LU, LUC and band factors were introduced into different models as independent variables for LSM analysis to verify the applicability of the three factors.

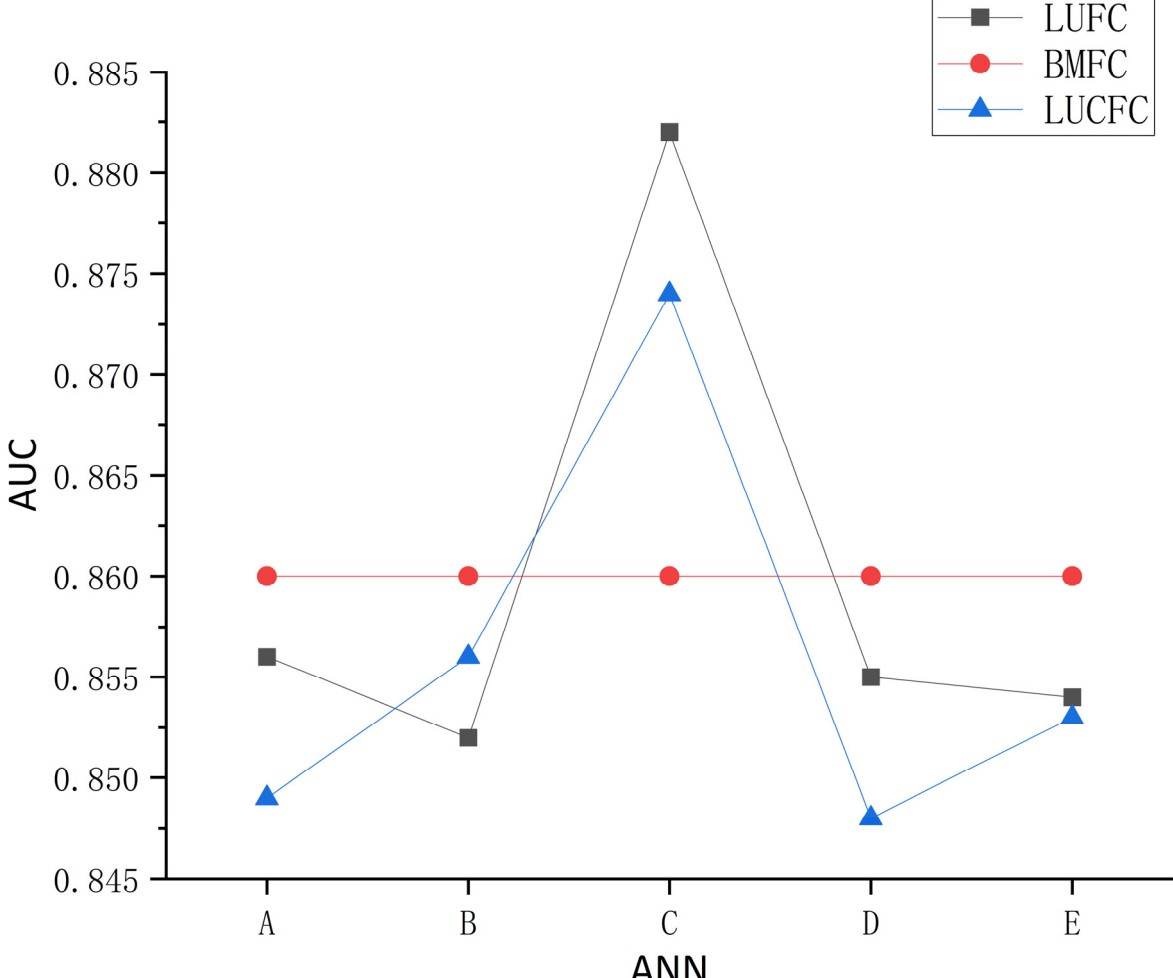

**Figure 17.** AUC values for the five groups of ANNs.

## 6. Conclusions

In this paper, the Zigui to Badong section was used as the study area, and the LUFC, LUCFC and BMFC were analyzed using three models. The ROC curve, specific category accuracy and five statistical methods (OA, precision, recall, F-measure and MCC) were used to evaluate the results of LSM. The predictive performance of each model was evaluated with a simple ranking method. The results showed that in general, the results of BMFC for the ANN model were better than those of LUFC and LUCFC, indicating that the band factor was more important than the LU and LUC factors in the three models. The simple ranking method verified that the score of BMFC for the three different models was higher than the scores of LUFC and LUCFC, indicating that the predictive ability of the band factor in the three different models was greater than that of the LU and LUC factors. Second, for the ROC curve analysis results (AUC values) for 2016 based on five groups of ANN experiments, as an example, the standard deviation of each factor combination was calculated, and the stability of BMFC was better than that of LUFC and LUCFC, indicating that the stability of the band factor was better than that of the LU and LUC factors. Therefore, compared with

the LU/LUC factors, with variations in accuracy, the band factor is better in principle and more stable.

Most existing landslide prediction models rely on human labor, which is limited by timeliness and accuracy, while machine learning methods can be used to accurately predict landslides in real time. According to the experimental results presented in this paper, especially the LSM results of the three different models, BMFC yielded better results than LUFC and LUCFC; that is, the band factor was better than the LU and LUC factors and can replace them to a certain extent. Moreover, since the LU and LUC factors are influenced by the subjectivity of the operator and are unstable, the corresponding prediction of landslides has some limitations. The stability factor band, obtained by introducing band math, not only results in better landslide predictions compared with those using the LU and LUC factors but also greatly saves time and labor and machine costs, providing theoretical support for automated landslide monitoring and the real-time evaluation of landslides.

**Author Contributions:** Writing—original draft preparation, X.Y. and Y.X.; writing—review and editing, X.Y. and Y.X.; visualization, Y.X. and W.J.; conceptualization, X.Y. and Y.X.; software, Y.X. and J.Z.; validation, Y.X. and J.Z.; formal analysis, X.Y. and J.Z.; investigation, Y.X. and J.Z.; resources, X.Y. and W.J.; data curation, W.J. and J.Z.; methodology, X.Y., Y.X. and W.J.; supervision, X.Y.; project administration, X.Y.; funding acquisition, X.Y. and W.J. All authors have read and agreed to the published version of the manuscript.

**Funding:** This study was supported by the National Natural Science Foundation of China (No.41807297), National Natural Science Foundation of China (No.42101375) and Innovation Demonstration Base of Ecological Environment Geotechnical and Ecological Restoration of Rivers and Lakes (No.2020EJB004).

**Institutional Review Board Statement:** Not applicable.

**Informed Consent Statement:** Not applicable.

**Data Availability Statement:** Remote sensing data and DEM data can be downloaded from public websites. However, basic geographic data, basic geological data and landslide distribution data are all confidential data in China. According to relevant regulations, these confidential data have been decrypted when we used them. Any researchers in related fields that need these decrypted data can contact the corresponding author.

**Acknowledgments:** We are grateful to the National Natural Science Foundation of China and Innovation Demonstration Base of Ecological Environment Geotechnical and Ecological Restoration of Rivers and Lakes. We are also thankful to the Headquarters of Prevention and Control of Geo-Hazards in the Area of the Three Gorges Reservoir for providing data and material.

**Conflicts of Interest:** The authors declare no conflict to interest.

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
