# Peer review of "Landslide Susceptibility Mapping Based on Multitemporal Remote Sensing Image Change Detection and Multiexponential Band Math"

_sustainability, doi:10.3390/su15032226_

Round 1

Reviewer 1 Report

please see the attached report

Reviewer 2 Report

This manuscript predict the Landslide susceptibility through the perspective of space and time. Congyi County of China is used to case study, as a whole, the research content of this manuscript relating to landslide susceptibility prediction is abundant. The idea of landslide environmental factors screening is interesting, this manuscript has some novelties in the area of regional landslide susceptibility prediction. However, there are some problems needed to be revised. I recommend a revision to this manuscript as follows:

(1) The research contents are not clearly reflected by the title of this paper.

(2) The abstract should be more concise. The study significance and novelty of this manuscript should be described in the abstract. Otherwise, how to identify the significance of your study? For example, the first sentence of the abstract should be deleted.

(3) The last paragraph of the introduction should be concise or be deleted. This is repeated in the “Methods” section below.

(4) Please simplify the section 2.2 and describe the section 2.3 in detail.

(5) Please Simplify section 2.5, you should not have four formulas in a pile

(6) Please explain how to classify the attribution numbers of each conditioning factors

(7) There are some English writing problem in this manuscript, please check the whole paper carefully.

(8) You should not use too many acronyms in a paper. 

(9) How practical is your proposed machine learning model for predicting landslide susceptibility? This needs to be analyzed in the discussion.

Reviewer 3 Report

This work uses deep learning method to provide landslide susceptibility mapping based on multitemporal remote sensing image change detection and multiexponential band math. It is an interesting work, and could be accepted after a minor revision.

1.     Section 5 is discussion, section 6 should be conclusion.

2.     In 3.3.3, the definition of recall, F-measure, and Matthews correlation coefficient should be given, rather than the formulars.

3.     Some references about the landslide hazard are suggested.

Ren, Z., Zhao, X., & Liu, H. (2019). Numerical study of the landslide tsunami in the South China Sea using Herschel-Bulkley rheological theory. Physics of Fluids, 31(5), 056601.

Li, L., Qiu, Q., Li, Z., & Zhang, P. (2022). Tsunami hazard assessment in the South China Sea: A review of recent progress and research gaps. Science China Earth Sciences, 1-27.

Round 2

Reviewer 1 Report

The necessary revisions have been penned and hence it can be accepted without further revision.

Author Response

Thanks for the reviewer's comments, which further improved this paper.

Reviewer 4 Report

There is not much improvement in writing. It should be either rejected or extensive English editing and correction is required. The authors can take help of native speakers to correct and improve the manuscript. Some of my comments are.

(i). The first sentence of introduction is too awkward.

(ii). The words like took in line 96 must be avoided.

(iii) line 81 and line 82: Why is there have in line 81 and has in 82? Do you want to say there are many CNN models and only one of them has good prediction ability?

(iv). First sentence of study area: why is [51] at the end?? Is it the reference for figure [2]??

(v) line 247- 249: “The acceleration of urbanization and the development of science, technology and the economy have further aggravated the occurrence of geological disasters in the study area for example, reservoir construction and expansion” does not look fine although it is clear what the authors mean to say. It should be “The urbanization and the infrastructure development activities have further aggravated the occurrence of geological disasters in the study area for example, reservoir construction and expansion”

(vi). There are other too many mistakes and awkward sentences for example under datasets “The digital elevation model (DEM) images were obtained through ASTER GDEM V2 data (https://asterweb.jpl.nasa.gov/gdem.asp) provided by National Aeronautics and Space Administration (NASA) and can generate altitude, slope, aspect and TWI.” What can generate altitude, slope, aspect, and TWI???? It can be simply written as “The DEM was obtained from ………… NASA. The obtained DEM can be used to generate altitude, slope, aspect, and TWI.”

(vii). From line 771 to 779: the information provided is not well written and it is confusing.

(viii) The first paragraph of conclusion section looks like discussion. The second paragraph looks better fit for the first paragraph of conclusion.

In summary, the writing is not good enough to be published. There are too many confusing sentences and mistakes which not possible to list here. The authors can take help from native speaker or from other agencies to correct the writing issues.

Round 3

Reviewer 4 Report

The manuscript is much improved now. Good luck